# Identification of Potential p38γ Inhibitors via In Silico Screening, In Vitro Bioassay and Molecular Dynamics Simulation Studies

**DOI:** 10.3390/ijms24087360

**Published:** 2023-04-17

**Authors:** Zixuan Cheng, Mrinal Bhave, Siaw San Hwang, Taufiq Rahman, Xavier Wezen Chee

**Affiliations:** 1School of Engineering and Science, Swinburne University of Technology Sarawak, Kuching 93350, Malaysia; 2Department of Chemistry and Biotechnology, Swinburne University of Technology, Melbourne, VIC 3122, Australia; 3Department of Pharmacology, University of Cambridge, Cambridge CB2 1PD, UK

**Keywords:** p38γ, QSAR modelling, virtual screening, molecular dynamic simulations, binding interaction

## Abstract

Protein kinase p38γ is an attractive target against cancer because it plays a pivotal role in cancer cell proliferation by phosphorylating the retinoblastoma tumour suppressor protein. Therefore, inhibition of p38γ with active small molecules represents an attractive alternative for developing anti-cancer drugs. In this work, we present a rigorous and systematic virtual screening framework to identify potential p38γ inhibitors against cancer. We combined the use of machine learning-based quantitative structure activity relationship modelling with conventional computer-aided drug discovery techniques, namely molecular docking and ligand-based methods, to identify potential p38γ inhibitors. The hit compounds were filtered using negative design techniques and then assessed for their binding stability with p38γ through molecular dynamics simulations. To this end, we identified a promising compound that inhibits p38γ activity at nanomolar concentrations and hepatocellular carcinoma cell growth in vitro in the low micromolar range. This hit compound could serve as a potential scaffold for further development of a potent p38γ inhibitor against cancer.

## 1. Introduction

Cancer is the second leading cause of non-communicable mortality worldwide, with an estimated 10.0 million cancer deaths in 2020 [1]. Cancer is often caused by dysregulation of protein kinases (either due to over-expression or constitutive activation) that triggers cancer cell proliferation and metastases [2]. Therefore, tumor-related kinases have emerged as one of the most important classes of drug targets in current cancer therapy [3]. To date, 58 small molecule kinase inhibitors (KIs) have been approved for the treatment of neoplasms by the US Food and Drug Administration [4]. However, the development of resistance, side effects, and compromised efficacy remain major limitations of KIs in the treatment of cancer [5]. Therefore, there is a pressing need to develop new inhibitors with high therapeutic efficacy against cancer.

The p38γ (MAPK12) is an isoform of p38 mitogen activated protein kinases (p38 MAPKs). Studies showed that p38γ is overexpressed in several types of primary human tumor cell lines, such as hepatocellular carcinoma (HCC), breast cancer, colorectal cancer, and osteosarcoma [6,7]. High expression of p38γ promotes the proliferation and migration/invasion of primary cancer cells by phosphorylating the retinoblastoma tumour suppressor protein (pRb) at the molecular level and stimulating the cell cycle protein cyclin E1/A expression [6]. Besides, the activation of p38γ induces G2/M phase arrest in cells and maintains cancer cell survival [8]. The knockdown of p38γ was shown to suppress cell growth and proliferation in cancer cell lines [9,10,11]. The lack of p38γ also attenuates the generation and development of cancer in mice [10,12,13]. These results strongly suggest that targeting p38γ could be an attractive strategy for developing cancer therapeutics.

However, to-date, pirfenidone is the only p38γ inhibitor approved for the clinical treatment of pulmonary fibrosis [14]. Pirfenidone was reported to exert anti-tumor effects mostly in vitro and/or in vivo by inhibiting p38γ activity [10,13,15,16]. Interestingly, studies showed that p38γ is associated with the intrinsic resistance of colon cancer to tyrosine KIs, and treatment with pirfenidone could restore drug sensitivity [8,17]. However, heavy doses of pirfenidone are required to reach inhibitory efficacy in clinical settings [17,18]. Meanwhile, PIK75, a phosphoinositide-3-kinase (PI3K) α-isoform inhibitor [19], was reported to be effective against p38γ for cutaneous T-cell lymphoma treatment [20]. However, PIK75 was also found to inhibit several other kinases [21]. Such binding promiscuity may lead to off-target toxicities. Therefore, the availability of p38γ inhibitors that are specific and selective is still an unmet need for cancer treatment.

In drug discovery, virtual screening (VS) has emerged as a core application in computer-aided drug design (CADD). VS techniques explore chemical libraries to identify novel hit molecules with expected activity through a variety of computational approaches [22]. Additionally, the advent of artificial intelligence (AI) further enhances the value of applying VS in drug discovery [23]. For example, the applications of predictive models with machine learning (ML) could accelerate drug discovery for the target of interest [24]. In this work, we attempted to identify potential p38γ inhibitors by screening the Enamine compound library through an extensive and systematic VS strategy and a biological evaluation (Appendix A). We developed an in silico drug discovery pipeline starting with machine-learning based quantitative structure activity relationship (QSAR) models complemented with conventional computer-aided drug discovery techniques (e.g., molecular docking and shape/electrostatic matching). The top hits were then subjected to 100 ns of molecular dynamics (MD) simulation and binding free energy calculations after filtering out undesirable pharmacokinetic properties and toxicity. Finally, the most promising hit compound was validated for its p38γ inhibitory activity and anti-proliferative effect in HCC cell lines by in vitro assays. As far as we are aware, this is the first report describing the in silico discovery of p38γ inhibitors.

## 2. Results and Discussion

### 2.1. QSAR Modelling to Identify Potential Hit Compounds

In any QSAR, the goal is to create a mathematical model that relates chemical features of compounds (e.g., molecular fingerprints) to their specific biological activity against a target (e.g., binding affinities) [25]. In this study, we obtained and cleaned a dataset from the ChEMBL database that consisted of the half-maximal inhibitory concentration (IC_50_) of various chemical compounds against p38γ. After cleaning, the final dataset comprised 103 compounds with negative log IC_50_ (pIC_50_) values ranging from 4.398 to 7.721. Next, we split the dataset into a training set of 92 compounds for building the QSAR regression models and an independent test set of 11 compounds for external test set validation. To make sure that the training and test datasets were congruent, we conducted principal component analysis (PCA) and showed that the chemical compounds in the training and test sets occupied the same chemical space (Appendix A). Next, we characterized the chemical compounds using three different types of two-dimensional (2D) bit vector molecular fingerprints [26] for subsequent comparison. These fingerprints are (a) the chemical development kit (CDK) extended fingerprint (with 1024 bits [27]), (b) the PubChem fingerprint (with 881 bits [28]), and (c) the Klekota-Roth fingerprint (with 4860 bits [29]). Each bit represents a molecular fragment as defined by the fingerprinting algorithm. Prior to ML modeling, we conducted feature selection to remove irrelevant molecular fingerprint bits to improve machine learning accuracy [30,31]. This yielded three dataset featurized with either 26 bits of CDK fingerprint, 13 bits of PubChem fingerprint or 13 bits of Klekota-Roth fingerprint, respectively. Subsequently, we used random forest (RF), support vector machines (SVM), radial basis function network (RBFNetwork), adaptive boosting (AdaBoost), multilayer perceptron (MLP), and k-nearest neighbor (kNN) to build our QSAR models. All in all, we constructed 18 ML-based QSAR models, and we tested their performance using 10-fold cross-validation (CV) and external test set validation (Appendix A) [32]. 

To select the best model, we assessed the accuracy of these models using the 10-fold cross validation method to determine the squared correlation coefficient (determination coefficients) ($Q_{10CV}^{2}$) and the mean absolute error (MAE) [33,34]. To this end, the top three models constructed were CDK-SVM, PubChem-SVM, and PubChem-AdaBoost as determined by their $Q_{10CV}^{2}$ (Figure 1) [35,36]. The $Q_{10CV}^{2}$ values of CDK-SVM, PubChem-SVM and PubChem-AdaBoost values were 0.75, 066 and 0.65, respectively. Across these three models, the SVM model with CDK fingerprint exhibited higher fitting accuracy with a lower MAE value (MAE = 0.48; Appendix A) compared to PubChem-SVM (*MAE* = 0.56; Appendix A) and PubChem-AdaBoost (MAE = 0.56; Appendix A) [33]. Taking both metrics into account, we determined that the SVM model with the CDK extended fingerprint provided a comparatively better overall performance. 

To make sure that our QSAR models were valid, we also validated our QSAR models using an independent external dataset and showed that the SVM-CDK model satisfies stringent rules developed by Tropsha et al. Based on the Golbraikh−Tropsha decision rule (GTR) [36], the CDK-SVM constructed QSAR model satisfied all of the criteria: the determination coefficient for the external test set ($R_{external}^{2}$) was 0.75, the determination coefficients for regressions through the origin (RTO; $R_{0}^{2}$ and ${R^{\prime }}_{0}^{2}$) were close to $R_{external}^{2}$, and the slopes of regression lines RTO (k and k′) were close to 1 (Table 1; Appendix A). At the same time, the concordance correlation coefficient (CCC), the conventional correlation-based external validation metrics ($Q_{Fn}^{2}$), and the mean value of the closeness between the validation and RTO determination coefficients ($\bar{r_{m}^{2}}$ )on the CDK-SVM model (Table 1) were higher than the proposed thresholds suggested by Chirico and Gramatica [37]. Moreover, our QSAR model provided an area under the receiver-operating characteristic curve (AUC) of 0.94 (Appendix A) when pIC_50_ = 6 was selected as the threshold to differentiate active and inactive compounds. Taking all of this information together, the data suggested that the CDK-SVM model was suitable for use to identify potential p38γ inhibitors.

Having identified a suitable QSAR model, we deployed our CDK-SVM model onto the Enamine compound database (~550,000 compounds). After screening using our QSAR models, 5006 hit compounds were predicted to inhibit p38γ with an IC_50_ less than 1 nM (pIC_50_ ≥ 6). 

### 2.2. Secondary Screening Using Molecular Docking and Ligand-Based Methods, and Follow-Up Negative Design to Shortlist Hit Compounds

Several studies showed that combining ML with conventional structure-based VS (SBVS) could enhance the hit rate [39,40]. Hence, these selected compounds from the QSAR model were further screened using molecular docking in our work. To improve the SBVS accuracy, we employed a two-stage docking protocol. The first stage of the docking uses AutoDock SMINA (SMINA) for rapid screening; the subsequent top hits were then screened again using the genetic optimization for ligand docking (GOLD) package.

First, we assessed the scoring functions in AutoDock SMINA (smina, vinardo, dkoes, and ad4) and GOLD (CHEMPLP, GoldScore, ChemScore, and ASP) for their suitability in our docking campaign through (a) cognate docking and (b) discriminatory ability on the p38γ dataset. In our cognate docking validation studies, docking protocols using all scoring functions were able to reliably reproduce the known crystallographic conformation except for SMINA-dkoes (Appendix A). This was determined by having their root mean square deviation (RMSD) values between the crystallized pose and docking pose (Table 2, Appendix A) be less than the threshold of 2 Å [41]. In our discriminatory ability studies, the scoring functions of SMINA-smina, SMINA-dkoes, GOLD-GoldScore and GOLD-ASP were able to distinguish active from inactive compounds well with AUC values (Table 2, Appendix A) greater than 0.7 [42]. We also attempted the use of consensus scoring to reduce false positives and improve hit rates in our VS [43]. We Z-normalized the docking scores from each protocol and combined them in three different manners, namely ASP+CHEMPLP, ASP+GoldScore, and ASP+CHEMPLP+GoldScore (Table 2, Appendix A). From here, the results of our validation studies supported the usage of SMINA-smina and GOLD-ASP+GoldScore as the tools for p38γ inhibitor screening. 

In our docking with SMINA, we used the docking score (−9.0 kcal/mol) of the ATP-competitive p38γ inhibitor PIK75 [20] as our benchmark. In total, only 1004 compounds had a docking score equal to or more negative than PIK75 and were selected for docking with GOLD. We then chose 500 of the top-ranked compounds from GOLD for subsequent study.

In parallel to the molecular docking studies, we also conducted three-dimensional (3D) ligand-based VS using the shape/electrostatic matching software Rapid Overlay of Chemical Structures (ROCS v3.4.1) and Electrostatic Similarity (EON v2.3.4) to screen the 5006 compounds shortlisted from our QSAR model. Before screening, we validated ROCS and EON using PIK75 as the query molecule. With PIK75 as the query molecule in ROCS, we achieved an AUC of 0.79 and an EF5% of 1.053 (Appendix A). Meanwhile, an AUC of 0.73 and EF5% of 0.351 (Appendix A). After sequential screening using ROCS followed by EON, we kept the 500 top-ranked compounds. 

Given the complementarity of molecular docking methods and ligand-based approaches, their combination could synergistically exploit the merits of these techniques and counterbalance their limitations [44]. Accordingly, we identified 70 hits shared in both the molecular docking and ligand-based screening by merging their respective top 500 hits (Figure 2A, Appendix A).

Further down the pipeline, the 70 hits were screened for their pharmacokinetic properties, frequent hitters, and toxicity (Figure 2B, Appendix A). From this step, two compounds were predicted to behave as pan assay interference compounds (PAINS), 11 compounds would aggregate in solution, and 44 compounds contained scaffolds that are known to have high promiscuity for various protein targets. No compound was reported with potential molecular toxicity. We used these predictions as a screening support to weed out the probable undesirable compounds. Altogether, 16 of the predicted compounds were kept (Appendix A). Among them, three hit compounds in sharing top-ranked compounds obtained from SMINA/GOLD docking and ROCS/EON matching were selected for MD evaluation: these compounds were Z1587203987, Z806431270, and Z2951708795, henceforth referred to as compound **1**, compound **2,** and compound **3**.

### 2.3. MD Simulation to Study Binding Stability of Hit Compounds

To evaluate the binding stability, 100 ns of MD simulations were performed for the top three compounds together with the positive controls pirfenidone and PIK75 (henceforth referred to collectively as ligands, Appendix A) in complex with in p38γ. 

The RMSD of a ligand-protein complex relative to its initial structure can provide direct insight into the binding stability of each system. As shown in Figure 3A, the RMSD fluctuations of all systems attained equilibrium after 60 ns simulation. There were no drastic RMSD fluctuations after equilibrium. This indicated the overall stability of these systems. The average RMSD values of the protein backbone and all ligands converged to within 3.1 Å and 2.5 Å, respectively, (Figure 4A, Appendix A). Furthermore, the RMSD values of the binding site were less than 2.0 Å (Figure 4B). These data suggest that the ligands were stable within the active site. Consistent with this analysis, we observed that the radius of gyration (Rg) fluctuation of these complexes reached equilibrium after 60 ns (Figure 3B). The average Rg deviation of all complexes were within 1.6 Å (Appendix A).

The effect of the ligands binding on p38γ was monitored by the flexibility/mobility of individual amino acid residues. The root mean square fluctuation (RMSF) values of backbone atoms in the complexes were measured throughout the simulations. The RMSF plot showed a similar pattern in the fluctuation of residues for all systems, including the pirfenidone- and PIK75-bound complexes (Figure 5A). The active site region of each complex fluctuated the least (less than 0.9 Å). Of particular interest was that Ile209 in the F-helix and Leu292 in the helical GHI subdomain showed different degrees of fluctuation depending on the type of ligands. 

Hydrogen bonding is essential for the overall stability of the ligand-protein structures. We selected 3.0 Å of the distance cutoff, 135° of the angle cutoff, and occupancy greater than 20% to analyze the hydrogen bond interactions of these simulation systems [45]. The hydrogen bond variations between the identified ligands and proteins during simulation are shown in Figure 5B. The result illustrated that all systems formed stable hydrogen bond interactions between hit ligands and p38γ throughout the simulation time. Compound **2** formed the most hydrogen bonds, followed by compound **1** and then compound **3**. In fact, the number of hydrogen bonds between compound **2** and p38γ was more than that of the pirfenidone- and PIK75-bound complexes. The binding poses of three ligands and PIK75 are shown in Figure 6.

To further evaluate the binding strength between the ligands and p38γ, the binding free energies of the simulation systems were investigated using the molecular mechanics generalized Born surface area (MM/GBSA) method. We decomposed by binding energetics into their respective components, namely the van der Waals contribution, the electrostatic energy, the electrostatic contribution to the solvation free energy, and the nonpolar contribution to the solvation free energy. The results are summarized in Table 3. The data showed that the binding of ligands to p38γ was mostly dominated by van der Waals force. This further reinforces the important role of hydrophobic interaction with p38 MAPK inhibitors [46]. The solvation provided unfavorable energy, possibly because of the larger binding pocket and the exposure of ligands to solvents. From our results, we noted that compound **2**-p38γ complex could bind with the most negative binding affinity (−38.99 kJ/mol) compared to compounds 1 and 3. Interestingly, the binding affinity energy was close to that of PIK75 and better than pirfenidone.

From here, we concluded that compound **2** exhibited good interactions with p38γ. However, in silico predictions are not meant to replace experimental determination. Therefore, we conducted an experimental evaluation of compound **2**. 

### 2.4. In Vitro Evaluation for p38γ Inhibitory Activity of Compound **2**

To validate the inhibitory effect of the most promising hit compound identified by our in silico strategy, the in vitro p38γ inhibition of compound **2** was first tested using p38γ kinase enzyme system with the ADP-Glo kinase assay. We evaluated the inhibitory activity against p38γ of compound **2** at concentrations of 1, 10 and 100 μM, respectively, and the pan-kinase inhibitor staurosporine was used as a reference control. The measurements of the enzyme activities indicated that compound **2** exhibited a higher efficacy compared to staurosporine at all three concentrations, and its inhibitory effects on p38γ activity are more pronounced at lower concentrations (Figure 7A). Compound **2** could inhibit p38γ kinase activity in a concentration dependent manner, which reduced p38γ activity by 72.43%, 93.12%, and 96.18% at 1, 10, and 100 μM, respectively. Additionally, compound **2** showed higher p38γ kinase inhibition compared to the known inhibitor pirfenidone (Figure 7A). Compound **2** was further tested at a wider concentration panel to generate a dose–response curve with a maximum inhibitor concentration of 100 μM in a four-fold 12-point serial dilution (Figure 7B). The dose-response curve of compound **2** on p38γ kinase activity gave an IC_50_ value of 16.80 nM. In comparison, the experimental IC_50_ for pirfenidone on p38γ was reported to be over 125 μM [20].

A lot of evidence showed that p38γ plays an important function in tumorigenesis and cancer development [6]. Therefore, p38γ inhibition is expected to reduce cancer cell proliferation. Therefore, we performed cytotoxicity assays with compound **2** to evaluate its antiproliferative activity against Huh7 and HepG2 cells. Both of these human HCC cell-lines were treated with three concentrations for 48 h. As reported in Figure 7C, compound **2** had no effect on the viability of Huh7 and HepG2 cells at a concentration of 1 µM. However, compound **2** displayed potent inhibitory effects on HCC cell growth at both concentrations of 10 and 100 µM. Compound **2** also showed equivalent inhibitory effects on Huh7 and HepG2 cells compared to sorafenib (a first-line therapeutic agent for HCC) at concentrations of 10 and 100 µM. Across all tested concentrations, compound **2** was far more potent than the p38γ inhibitor pirfenidone in the efficacy of inhibiting the proliferation of two HCC cells at these indicated concentrations. The growth inhibition of both Huh7 and HepG2 cells was also assessed through a dose-response curve with compound **2** in 10 concentrations performed three-fold serial dilutions of 100 μM (Figure 7D). The IC_50_ values of compound **2** in Huh7 and HepG2 cells were 11.87 μM and 19.24 μM, respectively. 

Overall, the results of the in vitro assays provided direct evidence that compound **2** is likely a p38γ inhibitor with anti-HCC activity.

### 2.5. Binding Analysis of Compound **2** to p38γ

Having confirmed the inhibitory activity of compound **2**, we then investigated the binding interactions between compound **2** and p38γ. To identify the key residues involved in the interaction of compound **2** with p38γ, the MMGBSA binding free energy of the complex was further decomposed into the per-residue contribution for proteins and small molecules. As shown in Figure 8A, Val41, Lys56, and Met112 showed the highest binding energy contributions with −2.35, −2.19, and −2.61 kJ/mol, respectively. This was followed by Phe111 with −1.90 kJ/mol and Leu170 with −1.80 kJ/mol (Figure 8A). Finally, other residues that contributed favorably to the binding affinities were the residues Val33, Tyr38, Asp115, and Ala160 (Figure 8A). The non-polar interactions take the dominant role (Figure 8B). Further energetic studies showed that Lys56, Met112, and Asp115 formed hydrogen bonds with compound **2** with a frequency of 84.70%, 76.46%, and 57.22% of the time in the 100 ns MD simulation (Figure 8C, Table 4). We also noted that compound **2** was also able to form hydrogen bonds with Met112 and Pro110, similar to that of the AMP-PNP bound with p38γ [47]. Throughout the simulation time, different hydrogen bonds were formed depending on the slight shift in binding conformation of compound **2** (Figure 9). 

To understand the effects of the proposed mode of action of compound **2** on p38γ, we compared the dynamic behavior of compound **2** in its bound state and in the apo state of the protein. The compound **2**-bound complex remained stable during the entire simulation after the 30 ns simulation mark. However, the apo protein continued to show higher fluctuations through the simulation (Appendix A). The fluctuations were even more pronounced when we examined the binding site RMSD plot of the compound **2** complex and the apo protein (Appendix A).

In order to further explore information on the conformational changes sampled by the protein in the metastable state over the simulation period, we also calculated the PCA and free energy landscape (FEL) of systems. The variances of the eigenvalues on the first 60 principal components showed that the compound **2**-p38γ complex had less correlated motion as compared to apo protein (Appendix A). The FEL along PC1 and PC2 was further analyzed (Appendix A). From the free energy surfaces (Figure 8D), we observed that apo protein explored a wide range of conformations with a single energy main basin. However, the compound **2**-p38γ complex sampled a more restricted conformational space and displayed two energy basins. These energy basins suggested that compound **2** was involved in the stabilization of the complex during simulation. This could mean that compound **2** potentially inhibits p38γ by preventing the protein from adopting its active conformation. This analysis was supported by the fact that the RMSF of the ligand-binding residues in the compound **2**-bound complex was less than that in the apo protein (Figure 8E). Interestingly, we noted that the fluctuations of the complex on residues 248–264 in MAPK-insert region were larger than those in the apo protein, pirfenidone, and PIK75 complexes (Figure 5A). The MAPK-insert plays the role of fine-tuning kinase activity [48,49] and is known to form a network of interdependent amino acids with the ATP-binding residues in the ligand binding site [50]. Moreover, the study had suggested that the changes in interaction of MAPK insert direct participants in p38γ activity [51]. Our MD simulation indicated that the binding of compound **2** could also affect the conformation change of the MAPK-insert. We hypothesized that this may indirectly regulate the activity of p38γ. 

Taken together, our analysis of compound **2** suggests that this compound could potentially regulate the activity of p38γ by trapping the protein in an inactive conformation or by dysregulating the MAPK-insert residues. 

## 3. Materials and Methods

### 3.1. QSAR Modelling

The dataset of the MAP kinase p38γ (Target ID CHEMBL4674) bioassays was extracted from the ChEMBL 27 database (https://www.ebi.ac.uk/chembl/, accessed on 17 February 2021). The activity entries with an affinity value reported as IC_50_ were collected. In cases of multiple experimental values for a compound, only one entry was kept for the median value. And the records with “>” “<” signs were kept only if they conformed to IC_50_ < 1000 nM or ≥1000 nM [52]. All compounds were converted to 3D structures using DataWarrior v05.02 [53] and OpenBabel v3.0.1 [54]. The CDK extended, PubChem, and Klekota-Roth fingerprints of compounds were respectively calculated using PaDEL-Descriptor 2.20 [55]. The dataset was then randomly split into a training set and a test set in 90:10. The feature selection of fingerprints was performed using Weka’s CfsSubsetEval attribute subset evaluators and the BestFirst search method [31]. 

Six ML algorithms (i.e., RF, SVM, RBFNetwork, AdaBoost, MLP, and KNN) were used to develop regression models for predicting the active compounds against p38γ:

RF is an ensemble learning method that creates a number of decision trees to evaluate data and performs predictions by selecting the best split point [56]. It fits multiple decision trees on various sub-samples of the dataset and uses voting or averaging to improve the predictive accuracy and avoid overfitting of the training data.

SVM is a supervised ML algorithm that establishes a hyperplane as the decision surface and works by finding the optimal hyperplane in the feature space for fit following the principle of structural risk minimization [57]. It has more flexibility in the choice of penalties and loss functions and scales better to large numbers of samples.

RBFNetwork is a group of instance-based methods that use the k-means clustering algorithm to provide the basis functions and learns either logistic regression or linear regression [58]. It implements Gaussian radial basis function networks for regression by minimizing squared error with the BFGS method and also is possible to be faster for cases with many parameters by using conjugate gradient descent.

Adaboost is a boosting algorithm that ensembles a number of simple predictive models (weak learners) to generate a comparatively strong model [59]. AdaBoost begins by fitting a regressor on the original dataset and then fits additional copies of the regressor on the same dataset, but subsequent weak learners are adjusted in an attempt to correct the errors of the previous learners.

MLP is an artificial neural network that can be constructed by organizing small processing units called neurons into different layers and connecting neurons to each other in consecutive layers [60]. MLP trains iteratively since at each time step the partial derivatives of the loss function with respect to the model parameters are computed to update the parameters.

KNN is an instance-based learning algorithm that works by storing the entire training dataset and querying it to locate the k most similar training patterns when making a prediction [61]. It is a simple algorithm that often achieves good performance, the only computation performed is the querying of the training dataset when a prediction is requested.

These models were built using Weka 3.8.5. Both 10-fold cross-validation and independent test set validations were used to evaluate the predictivity of all models. The internal validation parameters were given by the evaluation module of Weka, and the external validation parameters were calculated using XternalValidationPlus Tool v1.2. 

### 3.2. Consensus Docking

Autodock SMINA 1.1.2 [62] and Genetic Optimization for Ligand Docking (GOLD 5.3.0) were used for the docking screening. The 3D X-ray crystal structure of p38γ complex (PDB ID: 1CM8) was downloaded from the RCSB PDB (https://www1.rcsb.org/ accessed on 25 April 2021). The structure was optimized using UCSF Chimera 1.15 [63]. The co-crystallized ligand was deleted to form the ATP cavity as receptor protein. The binding pocket was detected on the ProteinsPlus server (https://proteins.plus, accessed on 30 September 2021). Ligands were energy minimized using the OpenBabel v3.0.1. For SMINA docking, the receptor protein was pre-processed using AutoDock Tools [64]. The grid size was delimited in xyz of 40 Å × 40 Å × 40 Å with center coordinates of 41.563, 75.213, and 4.552. The RMSD of 1.0 Å was used as the threshold of docking to filter final poses. Nine binding poses were generated for each ligand. The compounds obtained by SMINA screening were then imported into GOLD Wizard for re-docking. The detected residue list in the pocket was defined as the binding site of the protein. Each ligand was set to save 10 poses. 

### 3.3. Shape/electrostatic Matching

In this process, OpenEye Scientific software 2020.2.0 was used for shape/electrostatic matching. Specifically, the Rapid Overlay of Chemical Structures (ROCS) v3.4.1 was used for identifying potentially active compounds by 3D molecular shape comparison. Subsequently, the electrostatic similarity (EON) v2.3.4 was used for electrostatic potential similarity matching for the ROCS output compounds. The identified p38γ inhibitor PIK75 was used to generate the shape query. The p38γ dataset was used for ROCS query validation.

### 3.4. Negative Design Approaches

The selected compounds were submitted to the FAF-Drugs3 web server (https://fafdrugs4.rpbs.univ-paris-diderot.fr, accessed on 24 May 2021) for prediction of key physicochemical properties and pharmacokinetic parameters. Pre-defined drug-like filters were implemented for screening. Moreover, frequent hitters were estimated using PAINS Remover (https://www.cbligand.org/PAINS, accessed on 24 May 2021), ChemAgg (http://admet.scbdd.com/ChemAGG, accessed on 24 May 2021), and a bioassay-data associative promiscuity pattern learning engine (Badapple, http://pasilla.health.unm.edu/tomcat/badapple, accessed on 26 May 2021), respectively. The predicted pan assay interference compounds (PAINS), aggregates, and promiscuous (pScore ≥ 300) were excluded. Prediction of toxicity was conducted using the ProTox-II web server (http://tox.charite.de/protox_II, accessed on 27 July 2021).

### 3.5. Molecular Dynamics Simulations

The Amber software suite Amber20 [65] was used to run conventional MD simulations, on OzSTAR supercomputer at Swinburne University of Technology. The docked structures of ligand complexed with p38γ were used as the initial conformation of the system. The ligands were parametrized using the general amber force field (GAFF) by Antechamber module available in AmberTools20. Amber force field FF14SB was called out to parameterize proteins. The entire ligand-protein complexes were solvated into a truncated octahedron box of water with a margin of 10 Å using TIP3P water model. The solvated systems were electrically neutralized by adding an appropriate amount of neutralizing counterions Na+ ions. In total, the compound 1, compound 2, compound 3, PIK75, and pirfenidone systems contained 11,563 water molecules and 40,330 atoms, 11,562 water molecules and 40,334 atoms, 11,540 water molecules, and 40,260 atoms, 11,562 water molecules and 40,319 atoms, and 11,574 water molecules, and 40,339 atoms, respectively.

The PMEMD engine of AMBER was used for performing MD simulations. Periodic boundary conditions were applied for the simulation systems. Throughout the simulation, the SHAKE algorithm was applied to constrain fast motion bonds involving hydrogen atoms to allow integration of the force equation at 2 fs. The particle mesh Ewald approach was introduced to calculate long-range electrostatic effects, and a cutoff distance of 10 Å was applied to correct short-range electrostatic interactions. The Langevin thermostat with a collision frequency of 1 ps^−1^ was used to control and equalize the temperature. The systems were optimized through two stages of energy minimization. Firstly, water and neutralizing ions were minimized with a 500 kcal/mol restraint force by 1000 steps of steepest descent and 1000 conjugate gradients. Secondly, the entire system underwent minimization without any restraints by 1000 cycles of the steepest descent and 4000 of the conjugate gradients. After heating to 300 K in 100 ps with 100 kcal/mol restraint force, the systems were equilibrated during 200 ps under NVT ensemble. And then equilibration was continued with 1 ns of NPT ensemble on 1 bar system pressure. Finally, each system was performed with 100 ns of MD production under the NPT ensemble. The MD trajectory information was saved every 2 ps.

### 3.6. MD Simulations Analysis

The trajectory post-processing and data analysis were performed using the CPPTRAJ program in AmberTools20. The binding free energy of systems was calculated by the MM/GBSA method. A total of 500 snapshots were sampled from the last 10 ns trajectories every 20 ps. Per-residue free energy decomposition was performed. The calculations of the conformational entropy were omitted in view of the higher computational cost, because the shape and size of ligands are similar in the systems of this study [66]. The principal components were obtained by solving the covariance matrix of atomic fluctuations. The overall global and translation motions were removed prior to generating the covariance matrix. The PCA was calculated using the Python front-end package pytraj 2.0.5. The FEL was constructed using the open-source Python package PyEMMA 2.5.10 [67].

### 3.7. Kinase Assays of p38γ

The inhibitory assay of p38γ was performed using the p38γ kinase enzyme system (Promega, Madison, WI, USA) with the ADP-Glo™ kinase assay kit (Promega, Madison, WI, USA). The kinase reactions were conducted in a total assay volume of 5 μL in a white 384-well plate. The kinase reaction mixture per well was composed using the reaction buffer supplemented with 50μM DTT and contained 50 ng p38γ, 1μg p38 substrate, 50 μM ATP, and the tested compound at the specified concentrations. Positive control wells receive 5% DMSO instead of 1 μL of the compound solution, and negative control wells are without both compound and enzyme in the reaction. After incubation of the plates at room temperature for 60 min, the reaction was stopped with 5 μL of ADP-Glo™ reagent for 40 min. Next, 10 µL of kinase detection reagent was added and incubated for another 30 min to convert the ADP to ATP. The ATP was then detected by measuring the luminescence using a Victor Nivo multimode microplate reader (PerkinElmer, Waltham, MA, USA) in an integration time of 1 s per well. The relative kinase activity was calculated as follows: $$\begin{array}{c} p38\gamma \;kinase\;activity\;\%\\ =\frac{Lumenescence\;of\;compound-luminescence\;of\;negative\;control\;}{Lumenescence\;of\;positive\;control-luminescence\;of\;negative\;control\;}\times 100\% \end{array}$$

### 3.8. Cytotoxicity Assays

The compounds were tested for HCC cell lines using the Cell Counting Kit-8 (CCK-8) (Absin, Shanghai, China). The human HCC cell lines Huh7 and HepG2 were cultured in high-glucose Dulbecco’s modified Eagle’s medium (DMEM) supplemented with 10% fetal bovine serum (FBS), 1% penicillin, and 1% penicillin/streptomycin at 37 °C with 5% CO_2_. Huh7 and HepG2 cells were seeded into white 96-well plates at 3 × 103 cells per well in 100 μL suspension and pre-incubated overnight. The cells were then treated with different compound concentrations and incubated for 48 h. Control wells just contained cell suspension, while blank wells were added with complete media. The effects on proliferation were determined by the addition of 10 µL CCK-8 reagent, followed by incubation for 1 h. Finally, the absorbance of the reaction was measured at 450 nm using a Victor Nivo Multimode Microplate Reader. Cell viability was calculated as follows: $$Cell\;viability\%=\frac{\;experimental\;well\;absorbance-blank\;well\;absorbance\;}{control\;well\;absorbance-blank\;well\;absorbance\;}\times 100\%$$

### 3.9. Statistical Analysis

All data were collected from triplicate experiments and presented as the mean ± standard deviation of the mean. Data were analyzed using Graph Prism v9.0.0.121. One sample t test was used to calculate the statistical significance of the differences. *p* < 0.05 was considered statistically significant. The four-parameter dose-response curves and IC_50_ values were determined by a non-linear regression curve fitting method.

## 4. Conclusions

In this study, we developed a rigorous computational drug discovery pipeline to discover potential p38γ inhibitors. We first constructed a ML-based QSAR model that shortlisted potential p38γ inhibitors. These inhibitors were then subjected to parallel VS using extensively validated molecular docking and ligand-based methods. Through negative design filtering, we identified 70 potential p38γ inhibitors, and we selected the top three compounds for MD analyses. Through this extensive and rigorous screening, compound **2**, i.e., 1-(1,1-dioxothiolan-3-yl)-4-[4-(1H-pyrrolo [2,3-b]pyridin-3-yl)piperidine-1-carbonyl]pyrrolidin-2-one (Enamine, Z806431270) was identified as a promising active compound against p38γ. By validation of kinase and cell assays, the hit compound exhibited potent inhibitory activity against both p38γ protein (16.80 nM of IC_50_) and proliferation of Huh7 and HepG2 cells (11.87 μM and 19.24 μM of IC_50_, respectively). Moreover, we identified the key residues involved in the binding of compound **2** with the protein and proposed a mechanism of action. Our work here provides a valuable computational blueprint for the rational discovery of novel p38γ inhibitors for cancer treatment.

## Figures and Tables

**Figure 1 ijms-24-07360-f001:**
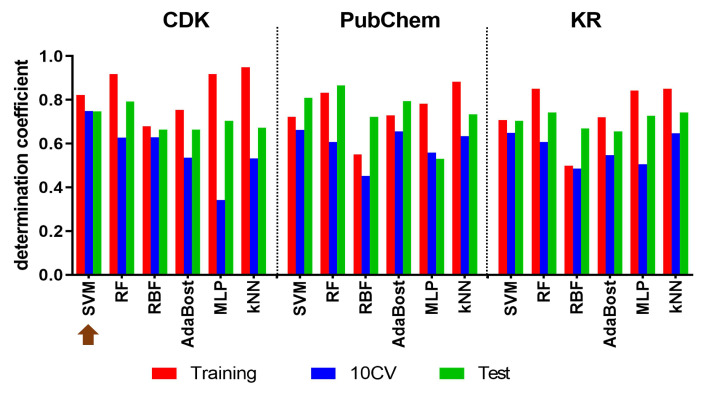
The determination coefficients from training set, 10-fold cross-validation and test set for each predictive modeling. The brown arrow represents the selected model for this QSAR-based screening.

**Figure 2 ijms-24-07360-f002:**
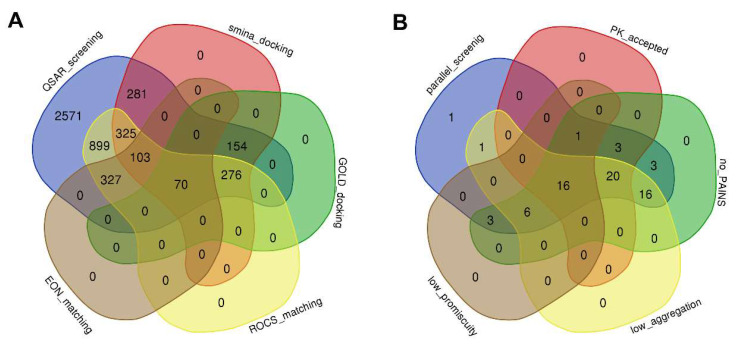
Venn diagram representation of (**A**) hit compounds yielded by QSAR-based screening, SMINA docking, GOLD docking, ROCS matching, and EON comparing; and (**B**) the compounds with desired properties kept by parallel screening, negative design, including physicochemical, PAINS, aggregates, and promiscuous filtering.

**Figure 3 ijms-24-07360-f003:**
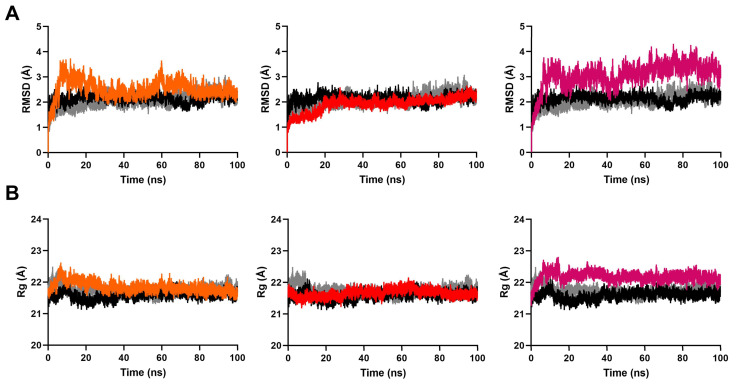
(**A**) Root mean square deviation (RMSD) plots and (**B**) radius of gyration (Rg) plots of the ligand-protein complexes were compared with controls during 100 ns MD simulations. In the figure, tangerine stands for compound **1**, red stands for compound **2**, fuchsia stands for compound **3**, black stands for PIK75, and grey stands for pirfenidone.

**Figure 4 ijms-24-07360-f004:**
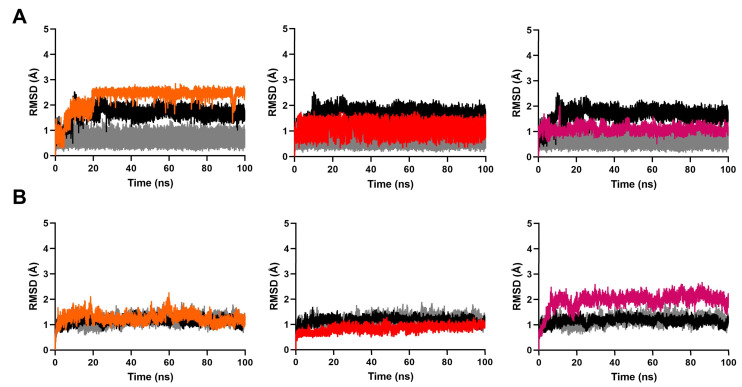
Root mean square deviation (RMSD) plots of (**A**) the ligands, and (**B**) the binding sites of the proteins obtained during 100 ns MD simulations. In the figure, tangerine stands for compound **1**, red stands for compound **2**, fuchsia stands for compound **3**, black stands for PIK-75, and grey stands for pirfenidone.

**Figure 5 ijms-24-07360-f005:**
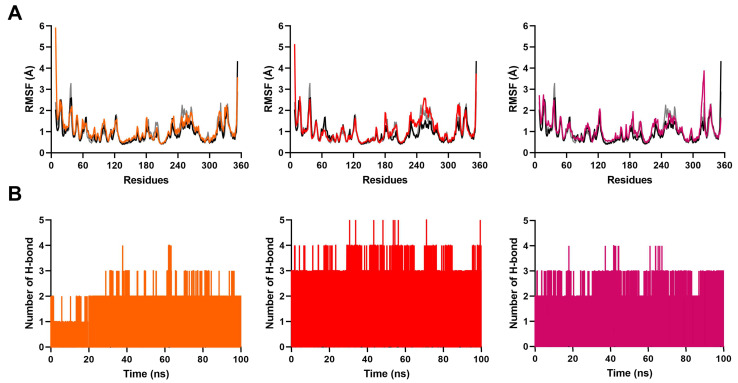
(**A**) Root mean square fluctuation (RMSF) plots of the ligand-protein complexes compared with controls during 100 ns MD simulations. (**B**) Number of hydrogen bond plots between ligands and proteins during 100 ns MD simulations. In the figure, tangerine stands for compound **1**, red stands for compound **2**, fuchsia stands for compound **3**, black stands for PIK75, and grey stands for pirfenidone.

**Figure 6 ijms-24-07360-f006:**
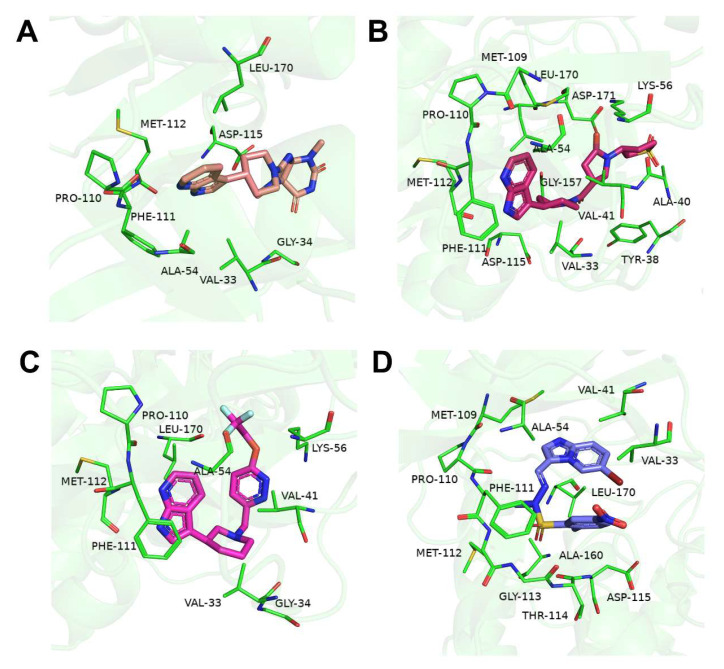
The binding poses of (**A**) compound **1**, (**B**) compound **2,** (**C**) compound **3**, and (**D**) PIK75 with p38γ at 100 ns simulations. The green lines represent the interaction residues of these ligands with receptor proteins.

**Figure 7 ijms-24-07360-f007:**
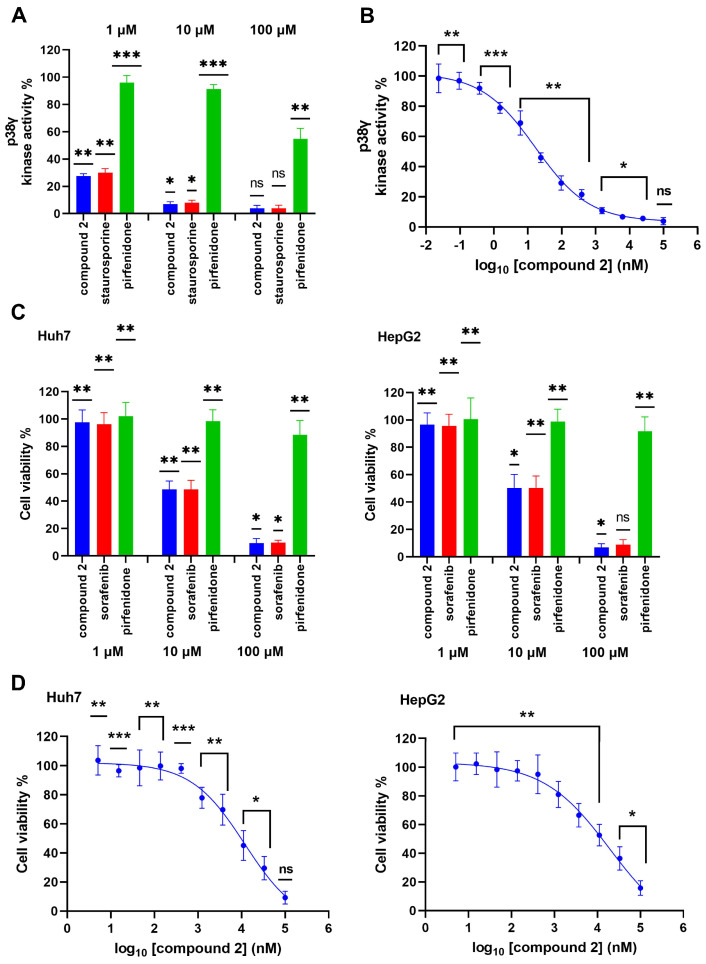
(**A**) Inhibition of p38γ kinase activity by compound **2**, staurosporine, and pirfenidone with indicated concentrations of 100, 10, and 1 μM. (**B**) Dose-response curve of compound **2** on p38γ kinase activity with a maximum concentration of 100 μM. (**C**) Inhibition of HCC cell viability by compound **2**, sorafenib, and pirfenidone for 48 h treatment at concentrations of 100, 10, and 1 µM. (**D**) Dose-response curves of compound **2** for HCC cells after 48 h of treatment. The data are represented as the mean ± SD of three independent experiments. ns *p* > 0.05, * *p* ≤ 0.05, ** *p* ≤ 0.01, and *** *p* ≤ 0.001.

**Figure 8 ijms-24-07360-f008:**
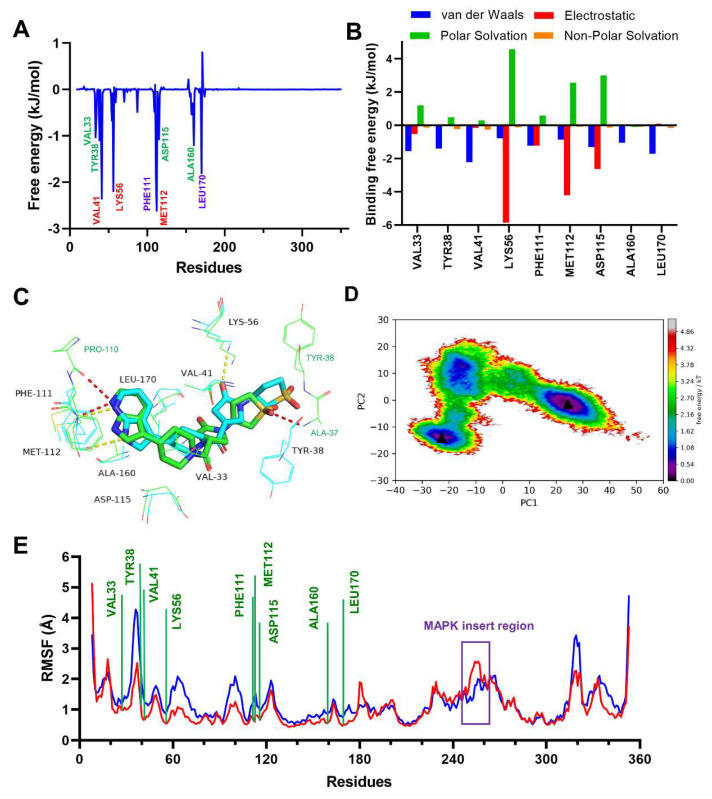
(**A**) Per-residue MM-GBSA binding free energy decomposition. (**B**) The binding free energy contribution of the key protein residues for the compound **2**-p38γ complex. (**C**) Superposition of compound **2** conformations in the docking structure (green) and the dynamic structure on the 100th MD (cyan). H-bonds from docking and MD are shown as red dashed lines and yellow dashed lines, respectively. The residues with green labels are marked for the docking structure. (**D**) Free energy landscape along the PC1 and PC2 of combined trajectory of compound **2** complex and apo. Lower left triangle represents the stand the coordinates of minimum free energy for compound **2** complex, right triangle right represents the coordinates of minimum free energy for apo. (**E**) RMSF plots of compound **2**-p38γ complex and apo protein.

**Figure 9 ijms-24-07360-f009:**
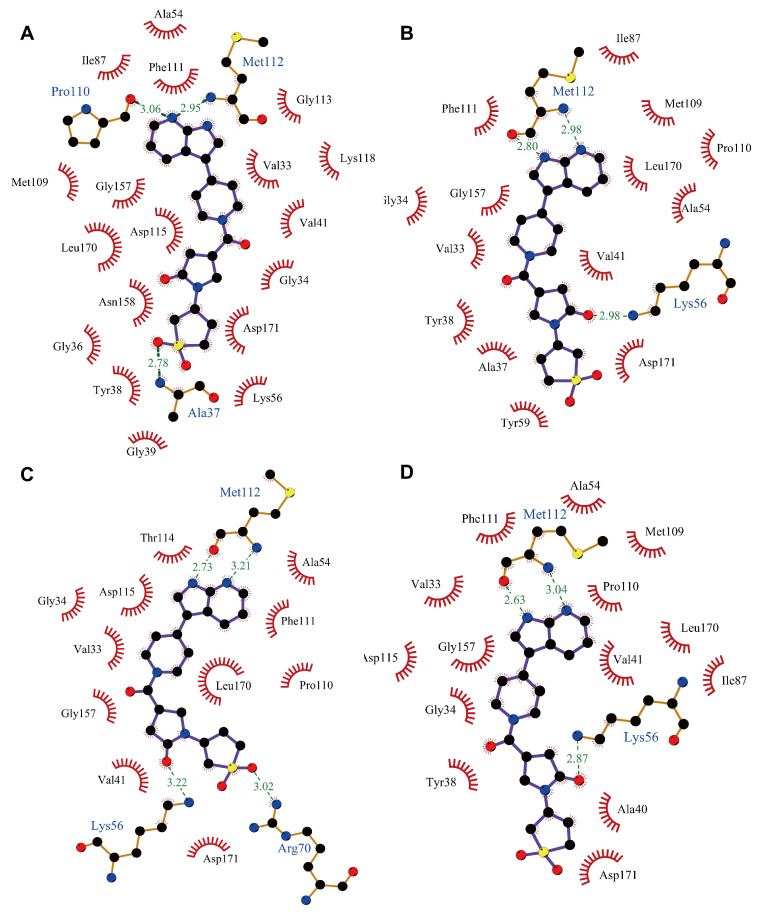
Hydrogen bond analysis of compound **2** with p38γ for (**A**) docking structure, (**B**) 20 ns snapshot, (**C**) 60 ns snapshot, and (**D**) 90 ns snapshot.

**Table 1 ijms-24-07360-t001:** The validation parameters of CDK-SVM model.

Metric	Equation	Proposed Threshold	Best Model Value	References
Cross validation				
Determination coefficient	$Q^{2}=1-\frac{\sum^{\;}{\left(Y_{i}-{\hat{Y}}_{i/i}\right)}^{2}}{\sum^{\;}{\left(Y_{i}-\bar{Y}\right)}^{2}}$	$Q^{2}$ > 0.5	$Q_{10cv}^{2}\;$ = 0.7491 $Q_{LOO}^{2}$ *=* 0.7600	[33]
External validation				
Determination coefficient	$R^{2}=1-\frac{\sum^{\;}{\left(Y_{i}-{\hat{Y}}_{i}\right)}^{2}}{\sum^{\;}{\left(Y_{i}-\bar{Y}\right)}^{2}}$	$R_{external}^{2}$ > 0.6	$R_{external}^{2}$ = 0.7475	[33]
Slope of lines for RTO	$k=\frac{\sum^{\;}Y_{i}\hat{Y_{i}}}{\sum^{\;}{\hat{Y_{i}}}^{2}}$ $k\prime =\frac{\sum^{\;}Y_{i}\hat{Y_{i}}}{\sum^{\;}Y_{i}{}^{2}}$	0.85 ≤ k ≤ 1.15 or 0.85 ≤ k′ ≤ 1.15	k = 0.9919 k′ = 1.0006	[33]
Determination coefficient for RTO	$R_{0}^{2}=1-\frac{\sum^{\;}{\left(Y_{i}-k{\hat{Y}}_{i}\right)}^{2}}{\sum^{\;}{\left(Y_{i}-\bar{Y}\right)}^{2}}$ ${R^{\prime }}_{0}^{2}=1-\frac{\sum^{\;}{\left(\hat{Y_{i}}-k\prime Y_{i}\right)}^{2}}{\sum^{\;}{\left({\hat{Y}}_{i}-\bar{\hat{Y}}\right)}^{2}}$	($R^{2}$−$R_{0}^{2}$)/$R^{2}$ < 0.1 or (*R*^2^−${R^{\prime }}_{0}^{2}$*)/*$R^{2}$ < 0.1 and $R_{0}^{2}$−${R^{\prime }}_{0}^{2}$ < 0.3	*(*$R^{2}$−$R_{0}^{2}$)/$R^{2}$ = 0.0026 ($R^{2}$−${R^{\prime }}_{0}^{2}$)/$R^{2}$ = 0.0719 $R_{0}^{2}$−${R^{\prime }}_{0}^{2}$ = 0.0518	[33,38]
Concordance correlation coefficient	CCC= $\frac{2\sum^{\;}\left(Y_{i}-\bar{Y}\right)\left({\hat{Y}}_{i}-\bar{\hat{Y}}\right)}{\sum^{\;}{\left(Y_{i}-\bar{Y}\right)}^{2}+\sum^{\;}{\left({\hat{Y}}_{i}-\bar{\hat{Y}}\right)}^{2}+n{\left(\bar{Y}-\bar{\hat{Y}}\right)}^{2}}$	CCC ≥ 0.85	CCC = 0.8604	[37]
$\mathrm{External}\;Q^{2}$ functions	$Q_{F1}^{2}=1-\frac{\sum^{\;}{\left(Y_{i}-{\hat{Y}}_{i}\right)}^{2}}{\sum^{\;}{\left(Y_{i}-{\bar{Y}}_{TR}\right)}^{2}}$ $Q_{F2}^{2}=1-\frac{\sum^{\;}{\left(Y_{i}-{\hat{Y}}_{i}\right)}^{2}}{\sum^{\;}{\left(Y_{i}-{\bar{Y}}_{EXT}\right)}^{2}}$ $Q_{F3}^{2}=1-\frac{\left[\sum_{i=1}^{n_{EXT}}{\left(Y_{i}-{\hat{Y}}_{i}\right)}^{2}\right]/n_{EXT}}{\left[\sum_{i=1}^{n_{TR}}{\left(Y_{TRi}-{\bar{Y}}_{TR}\right)}^{2}\right]/n_{TR}}$	$Q_{Fn}^{2}$ ≥ 0.70	$Q_{F1}^{2}$*=* 0.7435 $Q_{F2}^{2}$ *=* 0.7431 $Q_{F3}^{2}$ *=* 0.7150	[37]
Regression function	$r_{m}^{2}=R^{2}\times \left(1-\sqrt{R^{2}-R_{0}^{2}}\right)$ ${r^{\prime }}_{m}^{2}=R^{2}\times \left(1-\sqrt{R^{2}-{R^{\prime }}_{0}^{2}}\right)$ $\bar{r_{m}^{2}}=\frac{\left(r_{m}^{2}+{r^{\prime }}_{m}^{2}\right)}{2}$	$\bar{r_{m}^{2}}$ ≥ 0.65	$\bar{r_{m}^{2}}$ = 0.6585	[37]

$Y_{i}$ = experimental Y responses; ${\hat{Y}}_{i}$ = predicted Y responses; ${\hat{Y}}_{i/i}$ = predicted Y responses calculated excluding the ith element from the model computation; $\hat{Y}$ = mean value of experimental Y responses; $\bar{\hat{Y}}$ = mean value of predicted Y responses; n = number of compounds; *TR* = training set; *EXT* = external test set.

**Table 2 ijms-24-07360-t002:** Calculation of RMSD, AUC and enrichment values.

	RMSD (Å)	AUC	EF (5%)
AutoDock SMINA			
smina	0.474	0.7372	0.351
dkoes	2.540	0.7712	0.000
vinardo	0.214	0.6850	1.053
ad4	1.477	0.6895	0.702
GOLD			
CHEMPLP	1.342	0.6794	1.754
GoldScore	0.983	0.7018	1.053
ChemScore	1.012	0.6100	1.053
ASP	1.393	0.7248	1.404
GOLD consensus scoring			
ASP+CHEMPLP		0.7067	1.053
ASP+GoldScore		0.7243	1.053
ASP+CHEMPLP+GoldScore		0.7052	1.053

**Table 3 ijms-24-07360-t003:** Binding free energies (kJ/mol) of ligand–protein complexes.

Compound	Compound 1	Compound 2	Compound 3	Pirfenidone	PIK75
VDWAALS ^1^	−34.12	−49.82	−39.24	−24.08	−49.64
EEL ^2^	−30.33	−34.69	−86.51	−12.25	−18.95
EGB ^3^	41.72	51.55	98.85	15.10	33.90
ESURF ^4^	−4.02	−6.03	−5.81	−3.41	−5.71
DELTA TOTAL	−26.76	−38.99	−32.71	−24.64	−40.39

^1^ VDWAALS: the van der Waals contribution from MM; ^2^ EEL: the electrostatic energy as calculated by the MM force field; ^3^ EGB: the electrostatic contribution to the solvation free energy; ^4^ ESURF: nonpolar contribution to the solvation free energy.

**Table 4 ijms-24-07360-t004:** Hydrogen bonds interactions between compound **2** and p38γ.

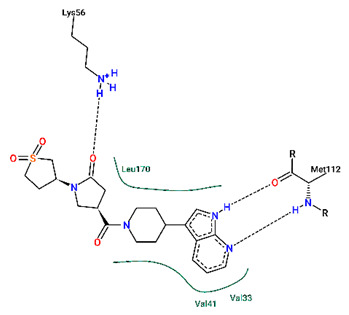	**Acceptor**	**Donor**	**Average Distance** **Å**	**Occupancy** **%**
	compound **2**_N2	p38γ_MET112_N	3.0976	76.456
	compound **2**_O1	p38γ_LYS56_N	2.8444	57.222
	p38γ_MET112_O	compound **2**_N1	2.8687	84.702

## Data Availability

Focused docking was conducted using the Genetic Optimisation for Ligand Docking (GOLD; version 5.3.0) using the free academic license courtesy of the Cambridge Crystallographic Data Centre (CCDC). All molecular dynamics simulation were conducted using the Assisted Model Building with Energy Refinement (AMBER; version 21.0) using a purchased academic license. The free platforms used in this study were presented in the descriptions of the methods. The dataset used during the current study is available from the corresponding author on reasonable request. All data and results generated during this study are included in this published article and the Appendix A.

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
