# Peer review of "Identification of Potential p38γ Inhibitors via In Silico Screening, In Vitro Bioassay and Molecular Dynamics Simulation Studies"

_ijms, 2023, doi:10.3390/ijms24087360_

Round 1
Reviewer 1 Report
The paper entitled “Identification of Potential p38γ Inhibitors via In Silico Screening, In Vitro Bioassay and Molecular Dynamics Simulation Studies” written by Cheng, Bhave, Hwang, Rahman and Chee presents combined studies performed to identify p38γ inhibitors against cancer. It is a combination of QSAR modeling, molecular docking and molecular dynamics simulations. Paper is interesting, However, IJMSc is intended for a wide audience. The work is written in a very hermetic language that is incomprehensible to a general audience. If the paper had been posted to a more specialist journal I would never address this point. However, the authors decided to send it to IJMSc that is traced by people from various fields. Therefore, they should make the paper more friendly. The development of tools that will identify inhibitors of given proteins and select the most promising - ones is of importance.
- each abbreviation and all quantities should be explained, e.g. IC50
- explanation concerning various chemical fingerprint should be given
- more details concerning MD simulations should be included (numbe of atoms, cutoff, etc)
- in my opinion all quantities used in the paper, especially those extensively used, should be defined by equations (fomal definitions)
Author Response
Point 1: Authors should make the paper more friendly.
Response 1: We thank the reviewer for the comment and we do apologize for the hermetic language. Wherever possible, we had edited the paper to make our contents clearer and more accessible.
Point 2: Each abbreviation and all quantities should be explained, e.g. IC50.
Response 2: We thank the reviewer for the comment. We have now defined the abbreviations at the first instance of their occurrences in the manuscript.
Point 3: Explanation concerning various chemical fingerprint should be given.
Response 3: We thank the reviewer for this comment. We had added in additional details about the chemical fingerprint (in lines 95-98) and the associated references should be readers want to know more in-depth.
Point 4: More details concerning MD simulations should be included (number of atoms, cutoff, etc).
Response 4: We thank the reviewer of the comment. We have now added additional details on how we conducted the MD simulations (in lines 509-520 at section 3.5. Molecular Dynamics Simulations).
Point 5: In my opinion all quantities used in the paper, especially those extensively used, should be defined by equations (fomal definitions).
Response 5: We agree wholeheartedly with the reviewer’s comment and we apologize for our oversight. We have added the equations of all the quantities in a newly created Table 1.

Reviewer 2 Report
This is an excellent study. The only faults that I can detect are insufficient descriptions of some of the methods. For instance, the sentence that begins on line 97 invokes several methods. As an example, the words Random Forest and the abbreviation RF do not seem to reappear in this work until Table S1. That fact demonstrates that the paper does not provide the resources to replicate this part of the work. This is not meant to suggest that a Random Forest procedure does not have utility in this work. I am quite sure it does. I only suggest that the procedures be more clearly explained.
Author Response
Point 1: insufficient descriptions of some of the methods.
Response 1: We thank the reviewer for the comment. We have now added more descriptions to our methods. These additional descriptions are for the algorithms that we used for modelling (in lines 433-462 at section 3.1. QSAR Modelling) and MD simulations (in lines 509-520 at section 3.5. Molecular Dynamics Simulations).

Reviewer 3 Report
In this manuscript, “Identification of Potential p38γ Inhibitors via In Silico Screening, In Vitro Bioassay and Molecular Dynamics Simulation Studies” by Cheng et al. a rigorous and systematic virtual screening framework to identify potential p38γ inhibitors against cancer. This manuscript is well written and could be accepted for publication.
Author Response
Point 1: This manuscript is well written and could be accepted for publication.
Response 1: We thank the reviewer for the kind comment.

Reviewer 4 Report
In this manuscript, the authors describe the "Identification of potential p38y inhibitors via In Silico Screening, In Vitro Bioassay and Molecular Dynamics Simulation Studies"
Although this manuscript is generally well written, it would be helpful if the authors address the following issues:
1) To prevent unnecessary distraction, it will be helpful if the description of figures are placed on the same page as the figure itself. A case in point is line 214 figure 3A and line 262 Table 3. Similar issues are also found in lines 333 - 337 for figure 7A.
2) In lines 217-218, referring readers to information in the supplemental section is counter productive, and certainly not helpful. It would be helpful if the authors provides those figures and tables (ie Figure S9A, Table S4 and Figure S9B) in the main manuscript.
Also, similar issues were found in line 340 (Figure 7C and Table S5) and line 344 (Figure S10). These are host of others throughout the manuscript. The authors should consider revising them all.
3) In line 268, the word is interestingly not interesting. Again, in line 282, the word is first not firstly. The authors should revise that.
In summary, this manuscript could potentially benefit it target audience if the above issues are addressed.
Author Response
Point 1: To prevent unnecessary distraction, it will be helpful if the description of figures are placed on the same page as the figure itself. A case in point is line 214 figure 3A and line 262 Table 3. Similar issues are also found in lines 333 - 337 for figure 7A.
Response 1: We thank the reviewer for the comment. We had changed the arrangement of Figure 3 and 4 so as to place the description of figure with the figure itself on the same page (lines 222, 240).
Point 2: In lines 217-218, referring readers to information in the supplemental section is counter productive, and certainly not helpful. It would be helpful if the authors provides those figures and tables (ie Figure S9A, Table S4 and Figure S9B) in the main manuscript.
Also, similar issues were found in line 340 (Figure 7C and Table S5) and line 344 (Figure S10). These are host of others throughout the manuscript. The authors should consider revising them all..
Response 2: We thank the reviewer for the comments. After reviewing, we have moved Figure S9, Table S5 and Figure S10 to the main text. These figures are now in the main text as Figure 4 (line 227), Table (line 381) and Figure 9 (line 384).
Wherever possible, we have moved the figures from supplementary materials to the main text as long as they do not obscure readership.
Point 3: In line 268, the word is interestingly not interesting. Again, in line 282, the word is first not firstly. The authors should revise that.
Response 3: We thank the reviewer for the comment. We have amended accordingly.
